# Resveratrol Reestablishes Mitochondrial Quality Control in Myocardial Ischemia/Reperfusion Injury through Sirt1/Sirt3-Mfn2-Parkin-PGC-1α Pathway

**DOI:** 10.3390/molecules27175545

**Published:** 2022-08-29

**Authors:** Minsi Zheng, Yinglu Bai, Xiuyu Sun, Rao Fu, Liya Liu, Mengsi Liu, Zhiyong Li, Xiulan Huang

**Affiliations:** 1School of Pharmacy, MINZU University of China, Beijing 100081, China; 2College of Life and Environmental Sciences, MINZU University of China, Beijing 100081, China; 3Institute of Chinese Materia Medica, China Academy of Chinese Medical Sciences, Beijing 100700, China

**Keywords:** autophagy, MIRI, mitochondrial quality control, resveratrol

## Abstract

Resveratrol is a natural polyphenol found in various plants. It has been widely studied on cardiovascular disorders. It is known that resveratrol can activate Sirtuin proteins and participate in cellular energy metabolism through a Sirtuin-dependent pathway. Here, we hypothesized that resveratrol may protect against myocardial ischemia/reperfusion injury (MIRI) through the target of Sirt1/Sirt3 on mitochondrial dynamics, cardiac autophagy, bioenergetics and oxidative damage in hypoxia/reoxygenation (H/R)-induced neonatal rat cardiomyocytes. We observed that resveratrol could activate the Sirt1/Sirt3-FoxO pathway on myocardial mitochondria in H/R cardiomyocytes. Subsequently, we found that resveratrol repaired the fission–fusion balance, autophagic flux and mitochondrial biosynthesis compared by H/R group. These changes were followed by increased functional mitochondrial number, mitochondrial bioenergetics and a better mitochondrial antioxidant enzyme system. Meanwhile, these effects were antagonized by co-treatment with Selisistat (Ex527), a Sirtuin inhibitor. Together, our findings uncover the potential contribution of resveratrol in reestablishing a mitochondrial quality control network with Parkin, Mfn2 and PGC-1α as the key nodes.

## 1. Introduction

Myocardial infarction (MI) is still the leading cause of morbidity, mortality, and economic burden worldwide. According to the 2019 Global Health Estimates released by the World Health Organization (WHO) in December 2020, ischemic heart disease accounts for 16% of all deaths in the world, and by 2019, its death toll has increased to more than 8.9 million. Clinical data show that the vast majority of myocardial infarctions are caused by coronary atherosclerosis. On the basis of coronary atherosclerotic stenosis, coronary atherosclerotic plaques are ruptured due to certain incentives, and platelets in the blood are on the surface of the ruptured plaques. Aggregation, and then the formation of thrombus, sudden blockage of the coronary lumen, and blocking of the coronary artery cannot provide the oxygen required for myocardial cell metabolism in time, eventually leading to myocardial cell ischemia necrosis and excessive apoptosis [1]. In addition, acute myocardial infarction can also be induced by a dramatic increase in myocardial oxygen consumption or coronary spasm. Common triggers are overworking, agitation, overeating, constipation, smoking, and heavy drinking. Restoration of myocardial perfusion, including percutaneous coronary intervention, thrombolysis, and coronary artery bypass grafting, is the most effective salvage method for myocardial ischemia [2]. However, these reperfusion measures may also lead to MIRI [3], including myocardial stunning, ventricular arrhythmias, and microvascular dysfunction [4], which possibly affect through a form of mitochondrial dysfunction. For example, the mitochondria are central to synthesis of both ATP and ROS, and mitochondrial and cytosolic Ca^2+^ overload are some of the main components of cell death [5,6].

Mitochondrial quality control is an important mechanism to ensure mitochondrial homeostasis and keep cardiomyocytes functionally viable [7]. Existing studies consider mitochondrial dysfunction as a hallmark behind the development of cardiac malfunction induced by MIRI [8]. In this regard, mitochondrial quality control, which targets the unhealthy mitochondria or aberrant mitochondrial proteins for degradation and removal, plays a vital role in maintaining myocardial homeostasis [9]. The intracellular mitochondrial quality control system includes multiple levels of regulation mechanisms, such as defense, repair, clearance and regeneration [10].

The study suggests that the NAD^+^-activated family of Sirtuin (Sirt) proteins is a receptor for intracellular nutrients and energy states, and has a tight connection with mitochondrial function [11]. To make it clearer, NAD^+^/NADH ratio is directly dependent on mitochondrial activity. What is more, Sirtuins act as histone deacetylases and regulate biological function, including cardiovascular related diseases [12].

The mitochondrial antioxidant enzyme system is the primary defense against mitochondrial damage, which can respond to ROS-mediated redox signals, achieve signal transduction, and maintain intracellular oxidation–antioxidant balance [13]. The molecular level of mitochondrial quality, as well as cytosolic proteolytic systems, selectively remove misfolded, redundant or damaged proteins in mitochondria while maintaining intact organelles and mitochondrial protein homeostasis [14]. Further, mitochondrial fission and fusion, mitochondrial autophagy and mitochondrial biosynthesis constitute the core of mitochondrial quality control [15,16]. Mitochondrial division and fusion are vital links in the dynamic control and repair of mitochondria. To put it more specifically, the PINK1/Parkin pathway may act synergistically to promote division by blocking fusion, thereby promoting mitophagy [17]. Mitochondrial autophagy and biosynthesis can promote the degradation and renewal of mitochondria, achieving dual regulation of the number and quality of mitochondria in cardiomyocytes. To make it simple, the mitochondrial fusion protein Mfn2 mediates Parkin recognition of damaged mitochondria in a PINK1-dependent manner, whereas PGC-1α, a crucial regulator of mitochondrial biosynthesis, regulates mitochondrial dynamics by regulating Mfn2 expression. In summary, we speculate that mitochondrial division and fusion, mitochondrial autophagy and mitochondrial biosynthesis with PGC-1α, Mfn2, and Parkin as key nodes form a multi-link regulatory network based on the promotion of mitochondrial quality, which together determine the “destiny” of cells.

With further study, more and more scientific evidence supports that healthy foods may be beneficial to human health. Resveratrol (3,4′,5-trihydroxystilbene; C_14_H_12_O_3_), a polyphenolic phytoalexin found in grapes, berries, peanuts, and wines, exhibits antioxidant, anti-inflammatory, anti-apoptotic, and anticancer capacities [18,19]. At nutritionally relevant concentrations, resveratrol increases antioxidative enzyme gene expression and inhibits transcription of proinflammatory cytokines [20,21]. Furthermore, resveratrol has recently been shown to improve MIRI [22]. Meanwhile, it was demonstrated that resveratrol enhanced mitochondrial biogenesis and mitochondrial function by activating Sirt1, subsequently enhancing AMPK activity. Moreover, a study has proved that Sirt3 in cooperation with Sirt1 activates FoxO, thereby promoting the activation of mitochondrial fission and mitophagy through the PINK1/Parkin pathway [23]. However, the effect of resveratrol on mitochondrial quality control processes in MIRI is unclear. Hence, the aim of the study was to investigate how H/R impairs mitochondrial quality control and later causes bioenergetic efficiency. What is more, we studied how resveratrol could improve cardiovascular fitness by modulating mitochondrial quality control processes, which provided an experimental basis for resveratrol against MIRI.

## 2. Results

### 2.1. Resveratrol Treatment Improves Stability of ΔΨm, ATP Content, SOD Content and MDA Content in Neonatal Rat Cardiomyocytes Undergoing H/R

To assess the degree of mitochondrial damage and effects of resveratrol on mitochondrial quality in H/R injured cardiomyocytes, we examined mitochondrial membrane potential (Δ*Ψm*), ATP content, the activity of SOD, and MDA content. First, cardiomyocytes upon resveratrol treatment resulted in an augment mitochondrial membrane potential (Δ*Ψm*) as demonstrated by JC-1 staining (Figure 1A,B). Paralleling our observations, resveratrol triggered a significant increase in the activity of SOD (Figure 1C). Moreover, the MDA content in the resveratrol treatment group showed a profound decrease compared with that in the H/R group (Figure 1D). While in the group that treated with Ex-527, a Sirtuin inhibitor, Δ*Ψm*, ATP and SOD levels were evidently reduced compared with those treated with 20 µM resveratrol, and MDA content increased distinctly from those of the Res-20 group (Figure 1A–D). Taken together, these results show that Sirtuin proteins are involved in the protective effects of resveratrol on mitochondrial anti-oxidizing effects and mitochondrial injuries induced by H/R.

### 2.2. Resveratrol Treatment Increases the Expression of Sirt1, Sirt3 and FoxO in Neonatal Rat Cardiomyocytes Suffered H/R

To investigate the effects of resveratrol on the regulation of Sirt1 and Sirt3 in cardiomyocytes injured by H/R, we detected the activities of Sirt1 and Sirt3 as well as the protein expression level of Sirt1, Sirt3 and their downstream proteins FoxO1 and FoxO3a. The deacetylase activity and the protein expression of Sirt1 and Sirt3 on the Res-20 group were significantly higher than that of the H/R group, and synchronously strongly higher than that of Ex527 group (Figure 2A,B). This finding was further supported with another observation where cells exposed to resveratrol responded with a slight increment in FoxO3a protein expression and a drastic increase in FoxO1 protein expression (Figure 2B). Moreover, the mRNA expression levels of FoxO1 and FoxO3a augmented significantly with increasing the dose of resveratrol, while the upregulations of the FoxO3a expression were significantly weakened by the Sirtuin inhibitor, Ex527 (Figure 2C).

### 2.3. Resveratrol Treatment Regulates Mitochondrial Function, Fission–Fusion and Biosynthesis in Neonatal Rat Cardiomyocytes after H/R

Mitochondrial dynamics, including mitochondrial fission–fusion and biosynthesis, are recognized as important constituents of cellular quality control and play a crucial role on the mitochondrial inheritance and on the maintenance of mitochondrial functions [24]. In this study we investigated the effect of resveratrol on mitochondrial dynamics in hypoxic/reoxygenation-induced cardiomyocytes and evaluated the role of Sirtuin activated in mitochondrial dynamics. We found that when mitochondria were stained with red and green fluorescent probes, the degree of the mitochondria network in the Res-20 group was greater than that in the H/R group. What is more, the number of functional mitochondria and the total number of mitochondria were both increased with the dose of resveratrol (Figure 3A). Moreover, in the resveratrol pretreated group, the mRNA expression levels of Mfn1, Mfn2, Drp1 and Opa1 were significantly increased with Fis1, and slightly increased as compared with the H/R group. These results illustrated that resveratrol could regulate mitochondrial dynamics by promoting mitochondrial fission–fusion and mitochondrial biosynthesis in cardiomyocytes treated by H/R. In the Ex527 group, the mRNA expressions of Drp1, Mfn1, Mfn2, and Opa1 in cardiomyocytes were inhibited compared with the Res-20 group; among them, the mRNA expression level of Mfn2 decreased more significantly (Figure 3B–D).

### 2.4. Resveratrol Treatment Improves the Neonatal Rat Cardiomyocytes Mitophagy Induced by H/R

To investigate if resveratrol also induces autophagy, we determined the expression and localization of mitochondria and lysosomes by using confocal fluorescent microscopy with MT-Green and LT-Red. Confocal fluorescent microscopic analysis showed a significant increase in the total number of mitochondria upon 20 µM resveratrol treatment, indicating that resveratrol could induce mitophagy. In addition, the number of functional mitochondria and the total number of mitochondria were obviously more attenuated in cardiomyocytes added by Ex527 than those in the Res-20 group (Figure 4A). What is more, the expression of LC3-I and LC3-II upon resveratrol treatment were analyzed by Western blot analysis in cardiomyocytes. Compared with the H/R group, the protein expression of LC3-II was significantly increased upon 20 µM resveratrol treatment (Figure 4B). To further validate the mechanism of how resveratrol regulates autophagy during H/R injury, we detected the expression of PINK1 and Parkin. PINK1 kinase activates the E3 ubiquitin ligase Parkin to induce selective autophagy of damaged mitochondria [25]. We found that application of 20 µM resveratrol significantly increased the level of Parkin (Figure 4B). In addition, we performed co-immunoprecipitation (Co-IP) to determine the Parkin/p62 interaction in cardiomyocytes. Co-immunoprecipitation (CO-IP) analysis revealed that Parkin strongly interacted with p62 in cardiomyocytes treated with resveratrol compared with that in the H/R group (Figure 4C). These results indicated that resveratrol could induce mitophagy via the Parkin-dependent pathway in cardiomyocytes injured by H/R. In the Ex527 group, Parkin was significantly decreased (*p* < 0.05), and the interaction between p62 and Parkin was weakened (*p* < 0.01), illustrating that mitophagy mediated by the Sirt1/Sirt3-Parkin pathway was an important mechanism of resveratrol regulating mitochondrial quality control in H/R cardiomyocytes.

## 3. Discussion

Given their pivotal role in ATP generation, redox balance, Ca^2+^ homeostasis and cell death, mitochondria have a direct relationship with MIRI. The maintenance of mitochondrial bioenergetics relies on the mitochondrial quality control that allows for the segregation of damaged mitochondria, which are further isolated and removed by autophagy, and further accompanying mitochondrial biosynthesis.

Here, using an in vitro model of H/R, we suggested that failing cardiomyocytes show disrupted mitochondrial quality control, characterized by loss of mitochondrial fusion–fission balance and reduced autophagic flux and biosynthesis. These changes were followed by increased mitochondrial fragmentation with reduced bioenergetic efficiency and excessive oxidative stress. Moreover, we demonstrated that resveratrol reestablished the balance of the fission–fusion of mitochondria, autophagic flux and mitochondrial biosynthesis in H/R cardiomyocytes, characterized by the increased efficiency of mitochondrial oxidative phosphorylation. Further, we supported an autophagic mechanism for resveratrol improvements in mitochondrial bioenergetics; inhibition of autophagic flux using Ex527 could abrogate the increased mitochondrial oxidative capacity triggered by resveratrol in a H/R model.

Sirtuins, a family of NAD^+^-dependent protein deacetylases regulate biological function, including cardiovascular related diseases [12]. At present, seven subtypes, including Sirt1~Sirt7 are found in mammals, all of which contain similar NAD^+^ binding and catalytic domain. However, their subcellular localization and tissue expression are different, making it a highly dispersed biological function in mammals [26,27]. For instance, FoxO protein is an important regulator of cell survival, proliferation and metabolism, and its activity is mainly regulated by Sirtuin-mediated deacetylation. Jacobs studied the interaction of Sirt3 with FoxO protein and found that Sirt3 overexpression can up-regulate the DNA binding activity of FoxO3a and promote FoxO3a-dependent gene expression, which is essential for regulating cellular ROS levels [28]. In our study, we found that resveratrol could up-regulate the deacetylase activity and protein expression of Sirt1 and Sirt3, which indicates that the Sirt1/Sirt3-FoxO pathway might be an important mechanism of resveratrol against mitochondrial H/R injury in cardiomyocytes.

Mitochondrial fusion–fission and mitophagy are important players of mitochondrial quality control in cardiac cells, which allows functionally impaired mitochondria to be rescued or eliminated upon metabolic stress [29,30]. The dynamic balance between fusion and fission determine the number, shape and distribution of individual mitochondria and mitochondrial networks, and meet the different physiological needs of cells [31,32]. Among the distinct multiprotein complexes, mitofusin 1(Mfn1) and mitofusin 2(Mfn2) tend to mediate extracellular membrane fusion through GTP-dependent dimerization. The main function of optic atrophy protein 1 (Opa1) is to maintain the stability of mitochondrial raft structure, which plays an important role in the remodeling of mitochondrial inner membrane [33]. Drp1, fission 1, Fis1, and mitochondrial fission factor (Mff) are mitochondrial outer membrane division regulators [34]. Measurement of Δ*ψ_m_* during single fusion and fission events demonstrates that fission may yield uneven mitochondria, where the depolarized ones are less likely to become involved in a subsequent fusion and are more likely to be targeted by autophagy.

Therefore, we next measured the expression and activity profile of the main GTPases involved in mitochondrial fusion and fission in the heart (Mfn1, Mfn2, OPA1, Drp1 and Fis1). The mitochondrial fission and fusion process is mainly completed by the GTPase of the macromolecular dynein superfamily mediated by mitochondrial membrane fission and fusion [34,35]. The mechanism of mitochondrial fusion is relatively complicated, and it needs to go through two links: mitochondrial outer membrane fusion and mitochondrial inner membrane fusion. Mfn1 and Mfn2 are the key molecules that regulate the fusion of the mitochondrial outer membrane. The cardiomyocytes treated with resveratrol could be observed to exert a partial beneficial effect by increasing the fission–fusion levels of mitochondria and reducing intracellular ROS levels, in parallel to the increase in ATP (Δ*ψ_m_* depolarization) in H/R cardiomyocytes relative to control cells. The expression level of Mfn2 mRNA in cardiomyocytes with Ex527 was significantly inhibited, indicating that Sirt1/Sirt3-Mfn2 may be an important pathway for resveratrol to regulate the dynamic changes of mitochondria mass.

Recent evidence demonstrates that activation of autophagy is critical to remove damaged mitochondria and protect the heart against acute and chronic stresses, such as MIRI [36]. Mitochondrial biosynthesis is considered to be an important compensation mechanism after mitochondrial autophagy degradation during MIRI to maintain cell mitochondrial homeostasis [37]. When myocardial ischemia happens, Parkin and FUN14 domain-containing protein 1(FUNDC1)-mediated mitochondrial autophagy can specifically eliminate dysfunctional mitochondria, reduce their digestion of ATP, and increase the threshold of open healthy mitochondrial permeability transition pore (mPTP) in cardiomyocytes, thereby exerting myocardial protection [38,39]. While during the reperfusion phase, the transient increase in oxygen concentration in cardiomyocytes leads to the explosive production of mitochondrial ROS, and the excessive opening of mPTP triggers apoptosis and causes irreversible damage of the myocardium. The possible explanation is that a large amount of carbon dioxide and bicarbonate produced by anaerobic respiration during ischemia could inhibit the occurrence of mitochondrial autophagy during reperfusion and aggravate myocardial damage [40]. In the present study, we found that moderately upregulated mitochondrial autophagy activity protects against ischemia/reperfusion injury to cardiomyocytes [41]. As the Sirtuin protein is a receptor for intracellular nutrients and energy states related to mitochondrial function, our study aims to reveal the regulation mechanism of Sirt1/Sirt3 and mitochondrial autophagy through the regulation of mitochondrial autophagy by resveratrol. In this experiment, we found that resveratrol increased the activation of LC3-II and the number of mitochondrial autophagosomes, indicating that resveratrol could induce autophagy, including mitochondrial autophagy in myocardial cells induced by hypoxia/reoxygenation.

Interestingly, Sirt1 regulates the deacetylation of FoxO1, which could raise autophagy activity in cardiomyocytes under starvation conditions and exert cardioprotective effects [42]. In addition, Sirt3 has made significant progress in the regulation of mitochondrial autophagy. Up-regulation of Sirt3 expression inhibits mitochondrial division by Drp1, reduces PINK1-mediated mitochondrial autophagy, and avoids cell dysfunction caused by massive mitochondrial degradation [28]. In order to explore the mechanisms through which resveratrol promotes autophagy, we examined the expression of PINK1 and Parkin. The PINK1-Parkin signaling pathway is a major pathway that mediates the Parkin-dependent pathway. When the mitochondria are depolarized, the input of PINK1 is blocked and accumulates at the outer membrane of the mitochondria. Meanwhile, Parkin in the cytoplasm is recruited to the mitochondria and attracts the autophagy receptor p62/SQSTM1 through its ubiquitin–protein ligase to initiate mitochondrial autophagy [25]. Mfn2 mediates Parkin recognition of damaged mitochondria and participates in the regulation of mitochondrial autophagy in a PINK1-dependent manner. Moreover, Mfn1, Mfn2 can be ubiquitinated in a Parkin-dependent manner [43], which could directly promote its rapid degradation by proteolytic enzymes, leading to increased mitochondrial division, which triggers mitochondrial autophagy [44]. Our result shows that resveratrol induces mitochondrial autophagy in myocardial cells induced by H/R through Parkin. Compared with cardiomyocytes with resveratrol, the expression of the Parkin protein in the Ex527 group was significantly decreased, indicating that the Sirt1/Sirt3-Parkin pathway is involved in the regulation of mitochondrial autophagy in cardiomyocytes by resveratrol. What is more, we found that manipulations in Fis1 and Drp1 expression levels were consistent with fission having a role in mitochondrial autophagy.

To further investigate the mechanism through which Sirt1/Sirt3-Parkin regulate mitochondrial autophagy, we determined the interaction between p62 and Parkin. The present study showed that pretreatment with resveratrol significantly enhanced the binding of p62 to Parkin, confirming the regulation of resveratrol on mitochondrial autophagy in hypoxic/reoxygenated injured cardiomyocytes. The addition of Ex527 impaired the interaction between p62 and Parkin in cardiomyocytes, which further indicates that Sirt1/Sirt3 activation up-regulates Parkin protein expression and promotes p62 interaction with Parkin, allowing damaged mitochondria to be recognized by autophagosomes and cleaned through mitochondria autophagy.

In conclusion, resveratrol can significantly increase the content of MMP, ATP and SOD activity in myocardial cells injured by hypoxia/reoxygenation, reduce the content of MDA in cells, and has a significant protective effect on the mitochondria of myocardial cells damaged by hypoxia/reoxygenation. The protective effect of resveratrol on mitochondria in cardiomyocytes from hypoxia/reoxygenation injury may be related to the activation of the Sirt1/Sirt3-FoxO signaling pathway. Resveratrol can promote mitochondrial fission and fusion through Sirt1/Sirt3, accelerate the fission of damaged mitochondria and mitochondrial network, promote material exchange between mitochondria, and regulate mitochondrial mass. The protective mechanism may be related to the up-regulation of mRNA levels of key regulators of mitochondrial fission and fusion mediated by Sirt1/Sirt3. We found an excessive accumulation of fragment and dysfunctional mitochondria in failing cardiomyocytes, most likely resulting from impaired mitochondrial fusion and defective autophagy and mitochondrial biosynthesis, which are three different mechanisms that coordinated to maintain intracellular mitochondrial quality. In addition, resveratrol reestablishes mitochondrial quality control and affects the function of healthy mitochondria. We propose that resveratrol improves the synergy between cardiac mitochondrial quality control and bioenergetic efficiency in MIRI though a Sirt1/Sirt3-Mfn2-Parkin-PGC1α pathway (Figure 5), which may contribute to a novel therapy against MIRI.

## 4. Methods

### 4.1. Reagents and Antibodies

Resveratrol (Figure 6) was purchased from Changsha Kanglong biologics Co., Ltd. (Changsha, China), dissolved in dimethyl sulfoxide (DMSO), and the final concentration of DMSO was 0.1%.

DMSO, 5-bromodeoxyuridine (5-BrdU) and Selisistat (Ex527) were obtained from Sigma (St. Louis, MO, USA). Dulbecco’s modified Eagle’s medium (DMEM)/F12, Hanks Balanced Salt Solution (HBSS), fetal bovine serum (FBS) and 0.25% trypsin were purchased from Hyclone (Logan, UT, USA). MitoTracker™ Red CM-H2XRos (M7513), MitoTracker™ Green FM (M7514), and LysoTracker Red DND-99 (L7528) were purchased from Invitrogen (Carlsbad, CA, USA). SOD detection kits were purchased from Dojindo (Shanghai, China). Assay kits for MDA and ATP were obtained from Nanjing Jiancheng Bioengineering Institute (Nanjing, China). Sirt1 and Sirt3 deacetylase fluorometric assay kits were obtained from Cyclex (Nagano, Japan). Fluorescent dye (JC-1) was obtained from Beyotime Biotechnology (Jiangsu, China). RIPA cell lysate was obtained from Biomiga (San Diego, CA, USA). The following polyclonal primary antibodies were used in this study: anti-LC3, anti-Beclin1, anti-Sirt1, anti-Sirt3, anti-FoxO1 and anti-FoxO3a (Danvers, MA, USA), anti-p62, anti-PINK1, anti-Parkin (Cambridgeshire, UK). FITC-conjugated anti-rabbit antibody was purchased from Millipore (Darmstadt, Germany). Reverse transcription kit was purchased from Takara (Tokyo, Japan). The qPCR kit was from Kapa Biosystems (Boston, MA, USA). All other chemicals used in this research were of analytical grade.

### 4.2. Cell Culture and H/R Treatment

Sprague–Dawley rats (1-day-old) were purchased from the Center of Experimental Animal in the Academy of Military Medical Sciences, China. All experimental animal procedures were approved by the Biological and Medical Ethics Review Committee of the Minzu University of China (1 September 2017) and followed the national ethical guidelines for laboratory animals in China. Spontaneously beating neonatal rat cardiomyocyte cultures were obtained from 1-day-old Sprague–Dawley rats as previously described [45]. To establish an in vitro model of H/R, the cardiomyocytes were cultured in glucose-free anoxic HBSS and transferred to a tri-gas incubator (Thermo Fisher, Waltham, MA, USA) at 37 °C in an atmosphere of 94% N_2_, 5% CO_2_ and 1% O_2_. After 12 h, the medium was replaced with fresh DMEM/F12 supplemented with 10% FBS, and the cardiomyocytes were cultured at 37 °C in the normoxic incubator (95% air and 5% CO_2_) for 12 h of reoxygenation.

### 4.3. Experimental Groups and Treatments

The neonatal primary rat cardiomyocytes that had been cultured for 96 h in a single layer with good synchronous beating were randomly and evenly divided into the following groups: (1) control group, cardiomyocytes were cultured for another 24 h under normal conditions; (2) H/In group R, cardiomyocytes were placed in DMEM/F12 medium containing 1% FBS, placed in a CO_2_ (37 °C, 5% CO_2_) incubator for 12 h, then hypoxia for 12 h, and reoxygenated for 12 h to establish hypoxic/reoxygenated myocardium; (3) 5 μM resveratrol group (Res-5 group): added DMEM/F12 medium containing 1% FBS, the final concentration was 5 μM resveratrol, and placed in CO_2_ (37 °C, 5% CO_2_) in the incubator for 12 h, and then treated with hypoxia for 12 h/reoxygenation for 12 h; (4) 20 μM resveratrol group (Res-20 group): DMEM/F12 medium containing 1% FBS. The cells were cultured in a CO_2_ (37 °C, 5% CO_2_) incubator for 12 h with a final concentration of 20 μM resveratrol, and then treated with hypoxia for 12 h/reoxygenation for 12 h; (5) in the Sirtuin inhibitor group (Ex527 group), 20 μM resveratrol and Ex527 (1 μM) were added 1 h before hypoxia and reoxygenation and incubated for 12 h.

### 4.4. Analysis of ΔΨm

For Δ*Ψ_m_* analysis, cardiomyocytes were harvested by using trypsin and washed by PBS prior to incubation with JC-1 (200 µM) for 20 min at 37 °C. The cells were pelleted through centrifugation and re-suspended in JC-1 staining buffer, and then detected by fluorescence microplate system (Molecular Devices, San Jose, CA, USA). Relative degree of mitochondrial polarization was quantified by measuring the red-shifted JC-1 aggregate, which were favored under conditions of high membrane potential with excitation at 490 nm and emission at 530 nm. Meanwhile, green-shifted JC-1 monomers tended to predominate under conditions of low membrane potential with excitation at 525 nm and emission at 590 nm. The Δ*Ψ_m_* was reflected by the ratio of red to green intensity of cells.

### 4.5. Measurement of Total Cellular ATP

ATP content in cardiomyocytes was measured by an ATP assay kit (Jiancheng Bio, Nanjing, China) following the manufacturer’s protocol. Briefly, cells were lysed by ultrasonic crushing in a hot bath, and ATP detection solution was mixed with luciferase solution. Bioluminescence was measured with a luminometer. ATP content was estimated according to a standard curve. The results were normalized to cellular protein concentration. Luminance was measured by a microplate reader at 636 nm (Promega, Madison, WI, USA).

### 4.6. SOD and MDA Assay

Cardiomyocytes were collected and ultrasonicated under ice bath conditions for cell lysis. The activity of SOD and content of MDA of the cardiomyocytes were measured according to the manufacturer’s instructions by a FlexStation 3 Multi-Mode Microplate Reader (Molecular Devices, San Jose, CA, USA) and normalized on protein content and expressed by normalizing on protein content.

### 4.7. Sirt1 and Sirt3 Deacetylation Assay

Deacetylase activity of Sirt1 and Sirt3 were measured by fluorometric assay kits (Cyclex, Nagoya, Japan) according to the manufacturer’s instruction. In this method, a reaction mixture containing fluoro-substrate peptide, lysyl-endopeptidase, NAD^+^, Sirt1 or Sirt3 assay buffer was prepared. The reaction was initiated by adding a protease inhibitor-free enzyme sample under thorough mixing. Fluorescence intensity was measured with excitation at 355 nm and emission at 460 nm while the reaction rate was kept constant.

### 4.8. Detection of Mitochondrial Quality Control and Dynamic Changes

The cells were incubated with MitoTracker Green FM (MTG)/MitoTracker Red CM-H2TM (MTR) dyeing work fluid in the normoxic incubator (95% air and 5% CO_2_). Five fields of view were randomly selected; the images were obtained by laser scanning confocal microscope (Leica TCS SP 5) and analyzed by using Image J (1.5.1, NIMH, Bethesda, MD, USA) software for evaluating the experimental results.

### 4.9. Detection of Mitochondrial Autophagy

The cells were collected with LYSR dyeing work fluid, and then MTG dyeing work fluid was added into the laser confocal small dish, incubating in the normal incubator (95% air and 5% CO_2_), respectively. Five fields of view were randomly selected; the images were obtained by laser scanning confocal microscope (Leica TCS SP 5) and analyzed by Image J software.

### 4.10. Co-Immunoprecipitation Assay

The cardiomyocytes lysates were pretreated with anti-Parkin antibody at 4 °C for 120 min, and then they were further incubated with protein G beads at 4 °C overnight. Cardiomyocytes proteins were extracted using SDS sample buffer and subjected to Western blotting analysis with the antibodies against Parkin, p62, β-actin, respectively, to evaluate the expression of autophagy-related proteins analysis.

### 4.11. Real-Time PCR

Total RNA was isolated from cardiomyocytes by using Trizol Regent (Life Technologies, Rockville, MD, USA) and quantified by using an NAS-99 spectrophotometer (ACTGene, Piscataway, NJ, USA) after primer design. The RNA integrity was detected by 1.2% agarose gel electrophoresis. Then, 2 μg of total RNA was added to each cDNA synthesis reaction by using a reverse transcription kit and a cDNA synthesis kit (Takara, Otsu, Shiga, Japan). The RNA mixture was taken into the QuantStudio 6 Flex Real-Time PCR system after the RNA mixing for amplification. Three duplicate wells were set for each group of PCR. Internal standardization prior to the gene expression of the control group used as reference, 2^−ΔΔCt^ method was utilized for relative quantitative analysis: ΔΔCt = (Ct_target gene_ − Ct_r-Actin_) test group − (Ct_target gene_ − Ct_r-Actin_) control group.

### 4.12. Western Blot Analysis

Total cell lysate from cultured cardiomyocytes were separated in SDS-PAGE and transferred to PVDF membrane and analyzed by Western blot. Briefly, samples were subjected to SDS-PAGE in polyacrylamide gels depending upon protein molecular weight. After electrophoresis, proteins were electro transferred onto nitrocellulose membranes. Equal gel loading and transfer efficiency were monitored using 0.5% Ponceau S staining of blot membrane. Blotted membrane was then blocked at room temperature and then incubated overnight at 4 °C with specific antibodies against β-actin (1:1000), Beclin1 (1:1000), p62 (1:500), Sirt1 (1:1000), Sirt3 (1:1000), FoxO1 (1:1000), FoxO3a (1:1000), LC3 (1:1000), PINK1 (1:1000) and Parkin (1:200), respectively. PVDF membranes were incubated with secondary antibody (1:10,000, Abgent, San Diego, CA, USA). Finally, ECL luminescent liquid was added to cover the membranes for Western blot analysis. Semi-quantitation of the target protein was undertaken by scanning densitometry in X-film using Image J software.

### 4.13. Statistical Analysis

Experimental data processing and statistical analysis were conducted with SPSS 22.0 software (SPSS Inc., Chicago, IL, USA) and presented as mean ± standard deviation ($\bar{x}\pm s$). Comparisons between groups were performed by the one-way ANOVA, the difference was considered statistically significant at *p*-value < 0.05.

## Figures and Tables

**Figure 1 molecules-27-05545-f001:**
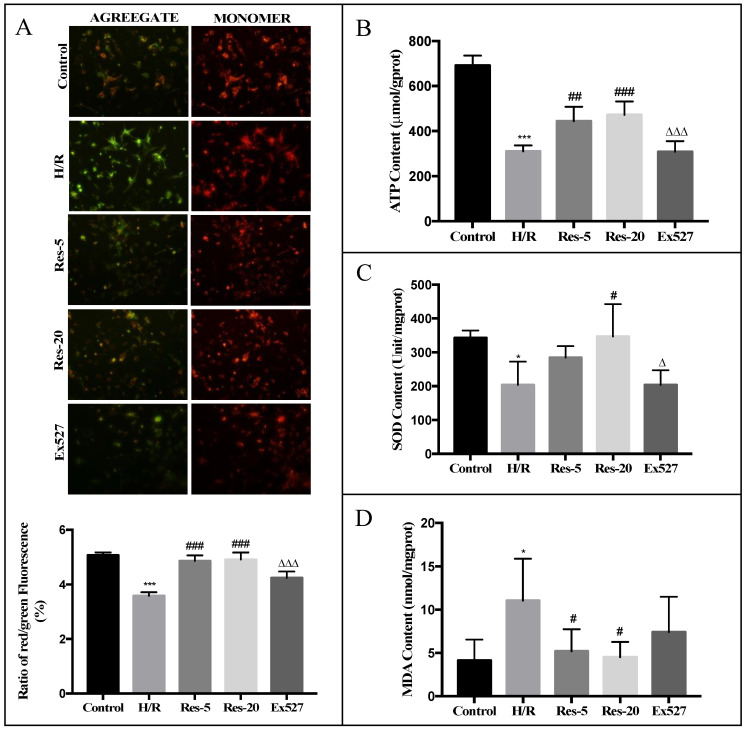
Resveratrol treatment improves stability of Δ*Ψm*, ATP content, SOD content and MDA content in neonatal rat cardiomyocytes undergoing H/R, while this effect was abrogated by EX527. The cardiomyocytes were pretreated with resveratrol at doses of 5 or 20 µM and resveratrol combined with Ex527 (1 µM) treatment exposed to hypoxia for 12 h and reoxygenation for 12 h. (**A**) Mitochondrial membrane potential (Δ*Ψm*) was determined by JC-1 dye, observed by inverted fluorescence microscope. Red aggregate emission profile represents the JC-1 probe accumulates in healthy mitochondria, whereas green emission represents mitochondrial injury and loss of membrane potential. The graph represents the ratio of red/green fluorescence. (**B**) ATP content was measured by inverted fluorescence microscope and microplate reader. (**C**) SOD content was measured by SOD detection kit-WST. (**D**) MDA content was measured by MDA test kit. Figures are representative images of three different samples. Results are expressed as mean ± S.D. * *p* < 0.05, *** *p* < 0.001 vs. control group; ^#^
*p* < 0.05, ^##^
*p* < 0.01, ^###^
*p* < 0.001 vs. H/R group; ^∆^
*p* < 0.05, ^∆∆∆^
*p* < 0.001 vs. Res-20 group.

**Figure 2 molecules-27-05545-f002:**
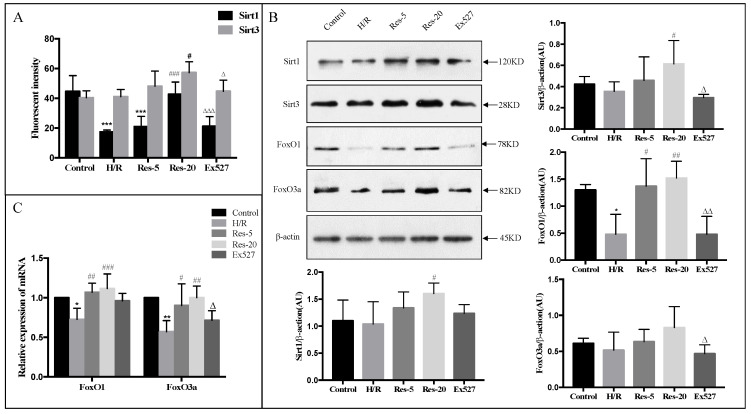
Resveratrol treatment increases the expression of Sirt1, Sirt3 and FoxO in neonatal rat cardiomyocytes suffering H/R, while this effect was abrogated by EX527. (**A**) Effects of resveratrol on Sirt1 and Sirt3 activities of H/R injury cardiomyocytes were measured by microplate reader. (**B**) Cardiomyocytes lysates were subjected to Western blot analysis for Sirt1, Sirt3, FoxO1 and FoxO3a. (**C**) The mRNA expression of FoxO1 and FoxO3a in cardiomyocytes was detected by Real-Time PCR. Figures are representative images of three different samples. Results are expressed as mean ± S.D. * *p* < 0.05, ** *p* < 0.01, *** *p* < 0.001 vs. control group; ^#^
*p* < 0.05, ^##^
*p* < 0.01, ^###^
*p* < 0.005 vs. H/R group; ^∆^
*p* < 0.05, ^∆∆^
*p* < 0.01, ^∆∆∆^
*p* < 0.001 vs. Res-20 group.

**Figure 3 molecules-27-05545-f003:**
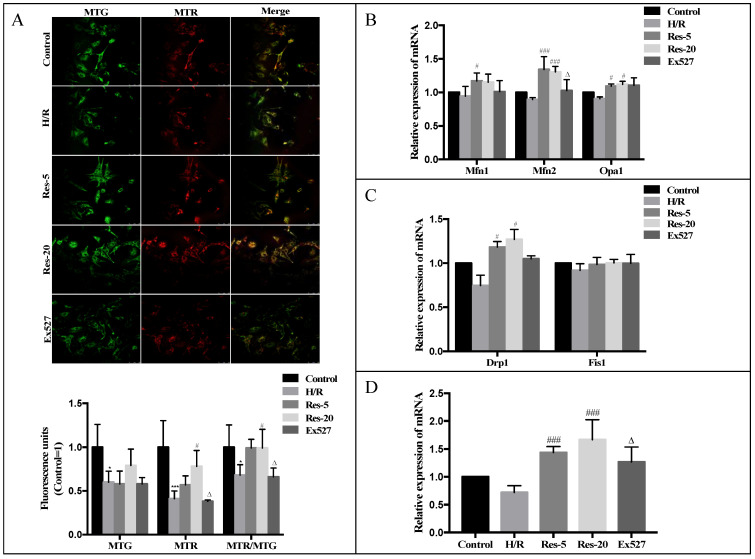
Resveratrol treatment regulates mitochondrial function, fission–fusion and biosynthesis in neonatal rat cardiomyocytes after H/R, while this effect was abrogated by EX527. (**A**) Representative picture of mitochondria probed with MTG and MTR through laser-scanning confocal microscopy. RT-PCR was performed with cardiomyocytes lysate for relative expression of (**B**) Mfn1, Mfn2 and Opa1, (**C**) Drp1 and Fis1, (**D**) PGC-1α. Figures are representative images of three different samples. Results are expressed as mean ± S.D. * *p* < 0.05, *** *p* < 0.001 vs. control group; ^#^
*p* < 0.05, ^###^
*p* < 0.005 vs. H/R group; ^∆^
*p* < 0.05 vs. Res-20 group.

**Figure 4 molecules-27-05545-f004:**
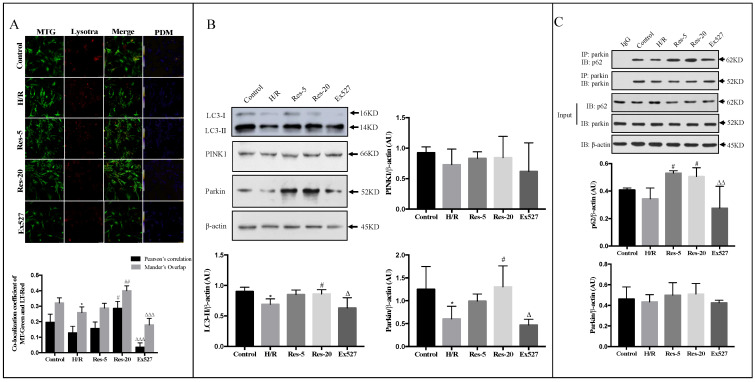
Resveratrol treatment improves the neonatal rat cardiomyocytes mitophagy induced by H/R, while this effect was abrogated by EX527. (**A**) Confocal fluorescent microscopic images of cardiomyocytes revealed the staining of mitochondrial autolysosomes. (**B**) Western blot was performed with cardiomyocytes lysate for expression of LC3, PINK1 and Parkin. (**C**) The interaction of Parkin with P62 was investigated by co-immunoprecipitation (Co-IP). Figures are representative images of three different samples. Results are expressed as mean ± S.D. * *p* < 0.05 vs. control group; ^#^
*p* < 0.05, ^##^
*p* < 0.01 vs. H/R group; ^∆^
*p* < 0.05, ^∆∆^
*p* < 0.01, ^∆∆∆^
*p* < 0.001 vs. Res-20 group.

**Figure 5 molecules-27-05545-f005:**
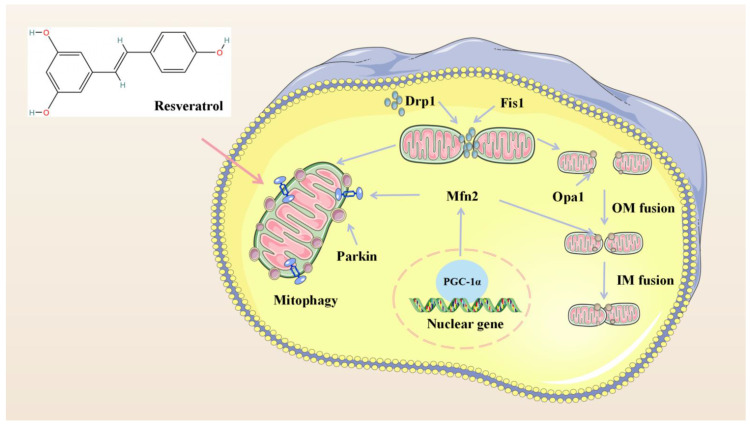
Overview of proposed mechanism of resveratrol reestablishes mitochondrial quality control in myocardial ischemia/reperfusion injury through the Sirt1/Sirt3-Mfn2-Parkin-PGC-1α pathway.

**Figure 6 molecules-27-05545-f006:**
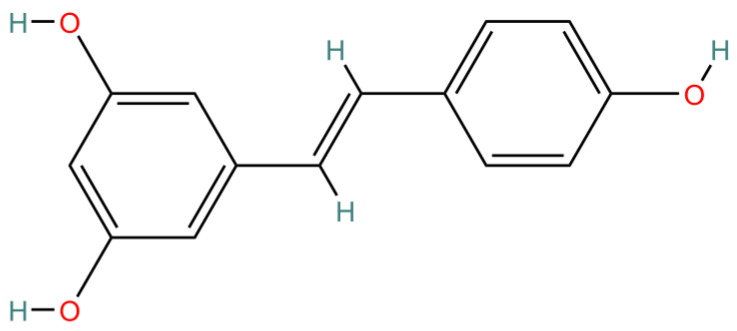
Resveratrol chemical structure.

## Data Availability

The data are available upon reasonable request from the authors.

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
