# Peer review of "Resveratrol Reestablishes Mitochondrial Quality Control in Myocardial Ischemia/Reperfusion Injury through Sirt1/Sirt3-Mfn2-Parkin-PGC-1α Pathway"

_molecules, 2022, doi:10.3390/molecules27175545_

Round 1

Reviewer 1 Report

Summary:

In the present paper, the authors present their research results that was carried out in order to demonstrate that resveratrol plays a crucial role in regulating mitochondrial quality control, and moreover this effect is produced involving Sirt1/Sirt3-Mfn2-3 Parkin-PGC-1α pathway.

The hypothesis is very interesting, the study is original and the study design is appropriate. The authors contribute to this field of research with new and valuable findings.

Nevertheless, several style and grammar revisions are recommended before publication.

Observations:

Line 8: Please correct the mistype: “plants,It has”

Line 27: “ischemia myocardia” should be “myocardial infarction”

Line 66: Please correct the mistype: “study, , more”

Line 66: health foods” should be “healthy foods”

Line 79: Please revise and rephrase the sentence: “…which provided pharmacological evidence for resveratrol’s prevention and treatment of MIRI.”

Lines 94 and 100: Please correct the following typos “Nanjing jiancheng bioengineering institute” and “Cambridge shire”

Line 138: Please specify exactly where the ATP kit comes from.

Line 160: Please describe precisely the used method. “Performed as the reference described,…” is not appropriate, especially since the reference is not specified.

Line 262: “an crucial” should be “a crucial”

Line 263: Please correct the reference

Line 299: Please revise and rephrase the sentence, as it can be misunderstood in this form: “We found that application of with 20µM resveratrol significantly increased the level of Parkin”

Line 305: “In Ex527 group,Parkin are significantly decreased” it should be “In Ex527 group, Parkin proteins are significantly decreased”

Line 357: Please rephrase the sentence to be more understandable: “Mfn1 and Mfn2, large GTPases that mediate mitochondrial fusion and mitophagy, accumulated in H/R cardiomyocytes.”

Lines 370 and 372: Please provide the meaning of the abbreviations “FUNDC1” and “mPTP”

Line 438: Please correct the mistype “Rao fu”

Line 455: The reference list does not meet to requirement of the journal Molecules.  The references must be numbered and in the text the numbers should be placed in brackets. See the "Instruction for authors" section.

Reviewer 2 Report

The paper entitled “Resveratrol reestablishes mitochondrial quality control in myocardial ischemia/reperfusion injury through Sirt1/Sirt3-Mfn2-3 Parkin-PGC-1α pathway” includes potentially relevant data for the pathobiochemistry of cardiovascular disorders as well as for the pharmaceutical and medical care focused on prophylactics myocardial infarction. The Authors of this publication indicated among others that natural polyphenol – resveratrol may actively regulate modulating the “events” participating in mitochondrial fusion protein Mfn2 determining Parkin recognition of damaged mitochondria in a PINK1-dependent manner.

Remarks:

1.     It is difficult to find in the “Introduction” subchapter information (even brief) on:

- Myocardial infarction risk factors and epidemiologic (present) data – it needs to be added; 

Resveratrol antioxidative mechanism of action – it needs to be added;

2.     It is not easy to find in the precise answers in the conclusion subchapter (even indirect) to the “scientific task” described in article aim.

3.     Figure no. 1 – the resolution is insufficient; structure is not accurately prepared (Authors can visit: https://pubchem.ncbi.nlm.nih.gov/compound/Resveratrol#section=2D-Structure); the figure description also needs to be remodeled – rather “chemical” than “molecular” structure.  

4.     The cited in the introduction subsection articles e.g. Murphy and Steenbergen, 2008Moens et al., 2005Gautier et al., 2008Leonard et al., 2003 do not represent the newest, available on e.g. medline/pubmed literature data.

5.     Some sections (Discussion – lines: 351-364, 381-386) lack of any citations. Therefore, it should be strongly emphasized, that in the mentioned section (Discussion) the achieved research results need to be confronted with a results of comparable studies. It must be added.  

6.     The Authors should precisely indicate / explain the use of the dose of resveratrol used in the experiment, given the lack of data on the standardization of the procedure described in the subsection “2.3. Experimental groups and treatments;

7.     The extremely information brief (the subsection "2.3. Experimental groups and treatments"), "…Control group, in which cardiomyocytes were incubated in normal conditions for additional 24 h…" (lines - 119-120)), makes impossible to assess the obtained results due to doubts on the correct selection/preparation of experimental controls – the indicated issue/problem must be explained.

Round 2

Reviewer 1 Report

Summary:

The manuscript was extensively revised by authors and was improved considerably, according to the recommendations.

However, some mistypes need to be corrected.

Please see below for the required fixes:

 Line 34: “millionClinical”

Line 139: “(2) H/ In group R,”

Reviewer 2 Report

The authors of the publication have modified the article.

The article - in its current form - is suitable for publication.